# Assembly of the Tripartite and RNA Condensates of the Respiratory Syncytial Virus Factory Proteins *In Vitro*: Role of the Transcription Antiterminator M_2-1_

**DOI:** 10.3390/v15061329

**Published:** 2023-06-06

**Authors:** Araceli Visentin, Nicolás Demitroff, Mariano Salgueiro, Silvia Susana Borkosky, Vladimir N. Uversky, Gabriela Camporeale, Gonzalo de Prat-Gay

**Affiliations:** 1Instituto Leloir, IIB-BA Conicet, Av. Patricias Argentinas 435, Buenos Aires 1405, Argentina; avisentin@leloir.org.ar (A.V.); ndemitroff@leloir.org.ar (N.D.); marianosalgueirounq@gmail.com (M.S.); sborkosky@leloir.org.ar (S.S.B.); gcamporeale@leloir.org.ar (G.C.); 2Department of Molecular Medicine and USF Health Byrd Alzheimer’s Research Institute, Morsani College of Medicine, University of South Florida, Tampa, FL 33612, USA; vuversky@usf.edu; 3Brazilian Biosciences National Laboratory, Brazilian Center for Research in Energy and Materials (CNPEM), Campinas 13083-970, SP, Brazil

**Keywords:** viral factories, condensates, LLPS, respiratory syncytial virus, Mononegavirales, M_2-1_, antiterminator, phosphoprotein, nucleocapsid

## Abstract

A wide variety of viruses replicate in liquid-like viral factories. Non-segmented negative stranded RNA viruses share a nucleoprotein (N) and a phosphoprotein (P) that together emerge as the main drivers of liquid–liquid phase separation. The respiratory syncytial virus includes the transcription antiterminator M_2-1_, which binds RNA and maximizes RNA transcriptase processivity. We recapitulate the assembly mechanism of condensates of the three proteins and the role played by RNA. M_2-1_ displays a strong propensity for condensation by itself and with RNA through the formation of electrostatically driven protein–RNA coacervates based on the amphiphilic behavior of M_2-1_ and finely tuned by stoichiometry. M_2-1_ incorporates into tripartite condensates with N and P, modulating their size through an interplay with P, where M_2-1_ is both client and modulator. RNA is incorporated into the tripartite condensates adopting a heterogeneous distribution, reminiscent of the M_2-1_-RNA IBAG granules within the viral factories. Ionic strength dependence indicates that M_2-1_ behaves differently in the protein phase as opposed to the protein–RNA phase, in line with the subcompartmentalization observed in viral factories. This work dissects the biochemical grounds for the formation and fate of the RSV condensates in vitro and provides clues to interrogate the mechanism under the highly complex infection context.

## 1. Introduction

Viral transcription and replication sites in RNA viruses are located at specific granular-like structures referred to as viral factories (VFs) that are formed within infected cells during viral replication. These structures are composed of viral proteins, nucleic acids, and host cell factors and serve as sites for viral replication and assembly of new virus particles [1]. Single negative-stranded RNA viruses (nsNSV) share the presence of cytosolic viral factories, with rabies virus (RABV) as a paradigmatic example [2,3,4]. Irrespective of the location of replication, nuclear in most DNA viruses or cytoplasmic for RNA viruses, a common feature observed during the course of infection is the formation of liquid-like structures to compartmentalize their gene function and particle assembly [5,6,7,8].

Human respiratory syncytial virus (RSV) and human metapneumovirus (HMPV) belong to the Pneumoviridae family (Mononegaviridae order); both are causes of acute lower respiratory tract infections and pneumonia mortality during the first year of life, especially in developing countries. Despite sequence divergence, these viruses share strikingly similar structural genomic organizations and transcription/replication mechanisms. The replication complexes of these viruses comprise an RNA-dependent RNA polymerase (L), a phosphoprotein (P), and a nucleoprotein (N). Replication requires newly synthesized RNA-free N protein (N0) to package the newly synthesized genomic and antigenomic RNAs. Transcription starts at the 3′ leader region of the 15 kb genome and proceeds through “gene start” (GS) and “gene end” (GE) sequences, yielding 10 individual transcripts encoding 11 proteins.

Unique to pneumovirus, M_2-1_ is an additional cofactor required for efficient mRNA transcription. Despite its key role in transcription, M_2-1_ can be deleted from the HMPV genome that still can replicate in cultured cells, although attenuated [9]. M_2-1_ is a 194 amino acid protein that functions as a transcription antiterminator protein and processivity factor, preventing the premature transcriptional termination of long mRNAs [10]. The X-ray crystallographic structures of M_2-1_ showed a tetrameric quaternary structure with three distinct regions linked by unstructured or flexible sequences [11]. It binds RNA sequences of 13 nucleotides or more with nanomolar affinity but no defined sequence specificity in the RNA binding [12]. In addition to participating in genome transcription, M_2-1_ is also associated with viral particle assembly, interacting in infected cells with P and N in the cytoplasmic inclusion bodies and, also, with viral mRNAs and cellular RNAs [13]. The NMR structure for the monomeric core domain of M_2-1_ confirmed the site for P binding [14].

RSV P is a homotetrameric protein of 241 aa involved in the replication and transcription of the viral genome. P interacts with N, L, and M_2-1_ [15,16], displaying a modular structure with three essential domains: the N0 binding intrinsically disordered domain, the oligomerization domain, and a C-terminal domain with a molten globule structure [17,18] and the N-RNA binding domain in a disordered stretch at the C-terminus [19]. The structure of the RNA polymerase L confirmed that the C-terminal domain of P binds with an unusual tentacular arrangement where the four C-terminal regions of each protomer adopt different conformations covering a large binding surface on the polymerase [20,21]. P contains multiple phosphorylation sites that regulate its function and interaction with other viral and host proteins [22]. 

RSV N is a 391 aa protein that consists of two globular domains, the N-terminal and *C*-terminal domains linked by a hinge region, conforming to a positively charged groove that binds RNA. Recombinant expression of RSV *N* yields decameric ring-like structures that bind nonspecifically to *ca.* 70 nt long RNA molecules from the host cell. N constitutes the coiled nucleocapsid and is recombinantly expressed as a highly stable decameric ring wrapped by RNA (N_R_ in this paper) [23].

The presence of dynamic protein and RNA containing granular structures was found to be based on the polymer chemistry principle known as liquid–liquid phase separation (LLPS) [24,25,26]. This phenomenon was observed to be the basis of a number of cellular substructures known as membraneless organelles. This term was coined later, giving place to biomolecular condensates (BMCs) [25]. Negri bodies (NBs) described in the early 1900s are characteristic structures found in nerve cells that have been infected with the rabies virus. NBs were shown to be involved in transcription, replication, and viral assembly and provided the first evidence that VFs are liquid-like organelles generated by LLPS, since soon after infection, NBs are small and spherical to fuse and become heterogeneous over time, and they can be dissolved by hypotonic shock [27]. 

The liquid nature of VFs refers to the fact that they are composed of a dynamic network of viral proteins and host cell components that are in constant motion. In the case of RSV, the cytosolic VFs contain L, N, P, M_2-1_, viral genomic RNA, and nascent viral mRNAs [28]. It was recently proposed that M_2-1_ and viral mRNAs transiently localize to VF subcompartments named IBAGs (IB associated granules) that exclude P, L, N, and genomic RNAs, whereby after IBAGs disassembly, M_2-1_ and mRNA would be released from VFs, suggesting that M_2-1_ drives viral mRNAs to the cytoplasm for translation [29]. In the case of Mononegavirales, the minimum requirement to form cytoplasmic liquid organelles is the interaction of N-RNA and P [30,31,32,33]. 

In this paper, we aim at dissecting the biochemical mechanism through which P-N- M_2-1_, as the main participants of the viral factories, assemble intro tripartite condensates, how RNA participates, and how these are modulated. For this, we made use of recombinant pure proteins, microscopy, and spectroscopic techniques to determine that M_2-1_ acts as an amphiphilic protein to condensate with RNA and form heterotypic condensates with P. Although N-P determines the formation of the condensates, M_2-1_ modulates the size of the condensates through interplay with P. We show that while electrostatic forces have little effect on the protein, only phase within the tripartite condensates, the highly electrostatic driven M_2-1_-RNA coacervate is responsible for partitioning these into heterogeneous subcompartments. We hypothesize that this mechanism could play a role in the formation and modulation of IBAGs and, therefore, the expression and trafficking of mRNA to the cytosol.

## 2. Materials and Methods

### 2.1. Protein Purification

All three proteins, N_R_, P, and M_2-1_, were expressed and purified as described in [23,34,35]. The protein concentration is expressed as a tetramer for P and M_2-1_, and as a decamer for N_R_. Far-UV circular dichroism, fluorescence spectra, and SDS-PAGE were performed to confirm the quality and purity of the proteins.

### 2.2. Protein Fluorescent Labeling

N_R_ and P proteins were labeled with fluorescein isothiocyanate (FITC) (Sigma-Aldrich, St. Louis, MO, USA), adapting the manufacturer’s protocol to obtain sub-stochiometric labeling enough to visualize the samples by fluorescence and confocal microscopy. A P:FITC molar ratio of 1:2 and a N:FITC molar ratio of 1:1 were used. Reactions were carried out at 4 °C overnight in sodium phosphate buffer 50 mM, pH 7.0. The reactions were stopped using 50 mM Tris-HCl pH 8.0 and excess FITC was removed by desalting PD10 columns (G & E), eluting each protein with the corresponding stock buffers.

A similar procedure was carried out for labeling P with DyLight-350 (Thermo Fisher Scientific, Waltham, MA, USA) using a P:DyLight-350 molar ratio of 1:0.25.

M_2-1_ protein was labeled with cyanine 5 (Cy5) maleimide or FITC maleimide (Lumiprobe, Hunt Valley, MD, USA), in phosphate buffer 50 mM, 300 mM NaCl pH 7.0, using a protein/fluorophore molar ratio of 2:1. Maleimide was used instead of isothiosyanate to avoid chemical modification of a lysine that affects the protein stability. Labeling procedure was similar to N_R_ and P.

These protocols yield proteins stocks labeled with 5–20% fluorescent dye. 

### 2.3. Turbidity Experiments

Turbidity was used as a qualitative parameter of condensation. Measurements were made at 370 nm using a Thermo Fisher Scientific Varioskan micro plate reader (Thermo Fisher Scientific, Waltham, MA, USA) for the microscopy experiments. All the conditions were measured in 96-well plates (Corning nonbinding surface). 

Complexes formed by M_2-1_-tRNA_70_ condensation were recorded in a Jasco UV spectrophotometer (Jasco, Hachioji, Tokyo, Japan) by following absorbance signals at 370 nm at 20 °C. Measurements were performed in 50 mM HEPES pH 7.5, 75 mM NaCl, and 5% PEG-4000. Addition of NaCl up to a concentration of 0.5 M was made after absorbance reached maximum signal. 

### 2.4. Bright Field and Fluorescence Microscopy

Imaging was performed on a Zeiss Axio Observer 3 inverted microscope (Zeiss, Oberkochen, Baden-Württemberg, Germany) using 40× and 100× objectives for brightfield and epifluorescence. Unless otherwise specified, samples were prepared in 25 mM HEPES buffer pH 7.5, 150 mM NaCl. The qualitative composition of condensates was determined by adding a labeled fraction of each protein (no more than 1 µM). The samples were loaded into 96-weel plates (Corning nonbinding surface) and incubated at room temperature for 3–4 h before observation. For fluorescence analysis, the samples were excited with LED light at 475 nm for FITC, 353 nm for DyLight 350, or 630 nm for Cy5. The images were acquired using an inverted Axio Observer 3 microscope (Zeiss, Oberkochen, Baden-Württemberg, Germany) with an LD A-Plan 40×/0.55 Ph2 objective. A Colibri 7 LED illumination system and a 90 HE LED filter set with BP 385/30-469/38-555/30-631/33 nm were used for excitation, QBS 405+493+575+653 nm as a beam splitter, and QBP 425/30+514/30+592/25+709/100 nm for emission. Acquired images were 1388 × 1038 pixels, with 0.233 μm × 0.233 μm pixel size, 16-bit, and cropped to 25 × 25, 50 × 50, 75 × 75, or 230 × 50 (µm). Images were processed using Fiji (a distribution package of the ImageJ software, National Institutes of Health, Bethesda, MD, USA).

### 2.5. Fluorescence Recovery after Photobleaching

Fluorescence recovery after photobleaching (FRAP) experiments were performed using a Zeiss LSM 880 Airyscan confocal laser scanning microscope (Zeiss, Oberkochen, Baden-Württemberg, Germany) with a C Plan-Apochromat 63×/1.4 Oil DIC M27 objective, 488 nm Argon Laser, and QUASAR detector at 495–571 nm.

Samples were incubated for at least 1 h on a Chamber Coverglass System with a BSA coating (Cellvis, *C8*-*1.5H*-*N* T) in 25 mM HEPES buffer pH 7.5, 75 mM NaCl, and 5% PEG-4000. A circular region of interest (ROI) with a one-third diameter relation of droplets settled at the bottom of the cover glass was bleached using 90% laser power. Droplets of an approximate diameter of 3 μm were selected for homotypic and heterotypic condensates, respectively (*n* = 7). A z-stack of three images was taken in two blocks of 90 cycles of 300 ms and 5 s, respectively, recording fluorescence intensity for three different ROIs (bleached droplet, reference droplet, and background). Acquired images were 196 × 196 pixels, with 0.09 μm × 0.09 μm pixel size, 16-bit, 3 slides with 0.8 μm of section, and 1.35 μsec pixel dwell. Fluorescence intensities from bleached ROI were corrected for photofading and normalized to the bleaching depth as described in [36]. Fluorescence recovery data were evaluated using the Fiji software (2.0.0-RC-68/1.52G).

### 2.6. Electro Mobility Shift Assay (EMSA)

Then, 2.5 µM of RNA_RSV20_ was incubated with increasing concentrations of M_2-1_ protein in binding buffer that contained 20 mM HEPES pH 7.5, 80 mM NaCl and 5% glycerol. All the samples were incubated at 20 °C for 20 min and loaded on a 4% polyacrylamide native gel as described in [37]. The electrophoresis was performed at 40 V for about 20 min. Visualization of RNA fluorescence was detected in a Storm Imager 840 (Molecular Dynamics, Amersham Pharmacia Biotech, Amersham, United Kingdom). Similarly, 1.5 µM tRNA_70_ was incubated with increasing concentrations of M_2-1_. A binding reaction was prepared in binding buffer that contained 20 mM HEPES pH 7, 0.1 M NaCl 5% glycerol, and 10 µM ZnCl_2_, as described in [34]. An electrophoretic run was made in an agarose 1.5% gel at 60 V for 1 h. tRNA_70_ and tRNA_70_-M_2-1_ complexes were visualized with ethidium bromide staining on a gel using a UV lamp. 

### 2.7. Partition Experiments

For determining the partitioning and stoichiometry of the protein components in the condensate, we mixed each protein with a fraction of labeled protein with either FITC, Cy5, or DyLight 350, according to the experiments. This allows us to modify protein concentrations keeping the fluorescence at a constant value. N, P, and M_2-1_ were mixed in 200 µL reaction volume at the indicated concentrations depicted in the results. A sample buffer consisted of 25 mM HEPES pH 7.5 and 0.075 M NaCl and the indicated PEG-4000 concentration (Appendix A). The samples were incubated for 1 h at 25 °C, followed by centrifugation at 13,000 rpm for 5 min, allowing the precipitation and direct visualization of the dense phase (in blue, corresponding to Cy5, Appendix A). Although not measurable accurately, we assume the total volume of the dense phase to be ≤2 µL, setting a lower limit for the concentration estimated. The dense phase was resuspended in 200 µL buffer without crowding agent and increasing NaCl concentration to 0.4M to ensure the dissolution of the condensates. The concentration of each protein was determined by fluorescence intensity of each fluorophore, measured in serial dilutions to confirm linearity. Each concentration is referred to the control where each protein was incubated in separate tubes in the absence of condensation. The concentrations determined in the condensed phase are an underestimation, as we set the residual volume to 2 µL, knowing it is lower but not accurately measurable.

### 2.8. Cell Imaging

A549 cells (human lung carcinoma, ATCC reference: CL-185) were grown in DMEM-F12K (Gibco, Grand Island, NY, USA) medium supplemented with 10% fetal bovine serum. For immunofluorescence experiments of transfected cells, 10^5^ A549 cells were co-transfected using Jetprime (Polyplus, Illkirch, France) with 0.3 µg of eukaryotic expression plasmids pcDNA-M_2-1_, -N, and -P fused with GFP at the N-terminus of the proteins. After 24 h, cells were fixed with 4% paraformaldehyde, permeabilized for 30 min with 0.1% triton-X 100 and blocked with 3% BSA in PBS. Cells were then incubated overnight at 4 °C with primary antibodies (mouse IgG anti-M_2-1_ RSV and rabbit IgG anti-N RSV), washed, and incubated for 1 h with goat anti-mouse Cy5- and goat anti-rabbit Cy3-conjugated IgG secondary antibodies (Jackson). Images were obtained on a Zeiss LSM510 confocal microscope (Zeiss, Oberkochen, Baden-Württemberg, Germany) using a Plan-Apochromat 63×/1.4 Oil immersion objective.

### 2.9. Bioinformatics Analysis of the Intrinsic Disorder Propensity

Predisposition for intrinsic disorder was evaluated by an intrinsic disorder profile generated using the outputs of the Rapid Intrinsic Disorder Analysis Online (RIDAO) platform [38] that collects the results from a number of well-known disorder predictors, such as PONDR^®^ VLXT [39], PONDR^®^ VL3 [40], PONDR^®^ VLS2 [41], and PONDR^®^ FIT [42], as well as IUPred2_Short and IUPred2_Long [43], and also produces the mean disorder profile (MDP) and distribution of corresponding standard. The outputs of the evaluation of the per-residue disorder propensity by these tools are represented as real numbers between 1 (ideal prediction of disorder) and 0 (ideal prediction of order). Thresholds of 0.5 and 0.15 were used to identify disordered and flexible residues and regions in query proteins. Residues with the disorder scores (DS) 0.15 ≤ DS < 0.5 are considered as flexible, whereas residues with DS ≥ 0.5 are considered as disordered. For each protein, the outputs of FuzDrop [44,45] are shown as the sequence distribution of the droplet-promoting probability. Thin gashed green lines correspond to the probability of droplet promotion, p_DP_, of 0.60, a threshold that is used to identify the droplet-promoting residues and regions (p_DP_ ≥ 0.60). Positions of the droplet promoting regions (DPRs) are shown by light green shadows.

### 2.10. RNAs

Single-chain RNA oligonucleotides were chemically synthesized and purified by the oligonucleotide synthesis service of the Brazilian Biosciences National Laboratory -CNPEM, Campinas, Brazil. RNAs labeled with 6-carboxyfluorescein (6-FAM) at their 5′ end were subjected to additional desalting and RP-HPLC purification. Quantification was made, using the corresponding molar extinction coefficients (ε260) by UV spectrophotometry to provide an accurate measure of concentration. The RNA sequences are the following (unlabeled RNAs lacked the 5′ 6-FAM moiety): 20-mer RNA: 5′-/6-FAM/AUAAAGUAGUUAAUUAAAAA-3′.

For the preparation of unfolded yeast tRNA, one milligram of sodium-tRNA (baker yeast tRNA, Sigma) was dissolved in 1 mL of milli-Q H_2_0. After the addition of 5 mM EDTA, the tRNA sample was incubated at 80 °C for 20 min and cooled at room temperature. Then, the tRNA sample was stepwise dialyzed against 10 mM sodium phosphate at pH 7.0 at 4 °C. The concentration of tRNA was determined based on the assumption that a 40 μg/mL solution had an absorbance of 1.0 at 260 nm. An average molecular weight of 25,000 g/mol was considered for the tRNA molecules to calculate the molar concentration. Thus, 40 μg/mL = 1.6 μM = 1 D.O_260_ nm.

## 3. Results

### 3.1. M_2-1_ Readily Undergoes Homotypic and Heterotypic Condensation with P

Phosphoproteins from Mononegavirales, including RSV, HMPV, and RABV, were shown to undergo homotypic LLPS to different extents [32]. In the case of RSV, we have shown that this tendency is driven by a molten globule interprotomeric domain downmodulated by intrinsically disordered regions adjacent to this domain [31]. As the starting point of this work, we addressed the formation of homotypic LLPS by M_2-1_ and found not only that it undergoes demixing but shows noticeably higher phase separation propensity than P. Under conditions of similar buffer and protein concentrations, M_2-1_ required 2.5% PEG as crowder, compared to the 15% PEG required by P (Figure 1A and [31]). In the presence of PEG and 150 mM NaCl, M_2-1_ displays a critical concentration (C_sat_) ≤ 0.5 µM compared to ≥6 µM for P (Figure 1B). Moreover, in the absence of crowder and at low ionic strength, M_2-1_ still displays LLPS propensity, but the shape of the condensates resembles a sticky bead-like morphology compared to larger and more regular droplets formed in the presence of 150 mM NaCl. This appears to respond to different physical properties of the condensates and requires further investigation. P does not form droplets in the absence of salt, even at 15% crowder (Figure 1B). We consider the C_sat_ of M_2-1_ to be low, ≤0.5 µM expressed as tetramer which is the stable species. Similarly, the most stable molecular species for N_R_ and P are decamer and tetramer, respectively [46].

Next, we tested the capacity of M_2-1_ to form co-condensates with P at different ratios. We found that both proteins coexist in droplets with a largely increased size as compared to droplets formed by M_2-1_ alone (Figure 2A). Based on the results of Figure 1B, we used an intermediate value of 75 mM NaCl for these experiments. Under similar conditions, M_2-1_ did not form condensates with N (Appendix A). Both M_2-1_ and P coexist in the droplets at a 1:1 ratio, but the size is minimal, something that could be based on both proteins forming a 1:1 stoichiometric complex with low nanomolar affinity, where M_2-1_ binds to a 9 amino-acid-binding site in the N-terminal domain of P [34,47]. A detailed concentration dependence analysis shows that an increase in P over a 0.25 P: 1 M_2-1_ ratio does not lead to a substantial increase in droplet size, whereas M_2-1_ yields a marked increase in the size of the heterotypic droplets (Figure 2B,C). We analyzed the tendency of heterotypic P-M_2-1_ mixtures to phase separate in a PEG concentration experiment and found it to be higher than that of the homotypic LLPS of M_2-1_, indicative of a mutually potentiated condensation (Figure 2D and Appendix A). 

The partition of the components between dilute and dense phases was investigated by quantitative fluorescence measurements of the protein concentration in each phase (Appendix A). We mixed either equimolar or excess ratios (4:1) of either of the components. Depending on the P:M_2-1_ ratios, both proteins concentrate between 5- to 30-fold in the dense phase, as the concentration in the dense phase corresponds to a minimal value (Table 1). Furthermore, at a 1:1 initial mixing ratio, we found a 1P:0.5M_2-1_ ratio in the dense phase (Figure 2E, Table 1). At a 4P:1M_2-1_ initial mixing ratio, the stoichiometry shifts to 16P:1M_2-1_ in the dense phase, while a 1P:4M_2-1_ initial mixing ratio yields a 1P:2M_2-1_ stoichiometric ratio in the condensate. We conclude that the stoichiometries in the condensate depend on the initial mixing ratios but also differ from them. P tends to be overpopulated in the condensate, suggesting a control on the stoichiometry, as well as a protein solvent role through homotypic interactions within the heterotypic condensate [31]. Homotypic LLPS of macromolecules arise in part from these having more binding propensity for themselves than for the actual solvent molecules, so scaffolds that are often in excess are acting in fact as solvents of the other components.

To evaluate the role of the N- and C-terminal domains of P on this heterotypic LLPS, we used the deletion of either of the domains, referred to as P_∆N_ (deletion 1–103) and P_∆C_ (deletion 161–241) [17]. At a 1P:4 M_2-1_ ratio, both P constructs form droplets that resemble those formed by the full-length P, consistent with interactions of M_2-1_ with both domains (Figure 2F), not restricted to the low nM affinity binding site at the N-terminal domain of P (aminoacids 93–110) [35], suggesting minimal sequence specificity, if any.

### 3.2. Formation and Modulation of Tripartite N_R_-P-M_2-1_ Condensates

Since in addition to the L polymerase, the main viral replication components are P, N_R_, and M_2-1_, we wanted to gradually build our understanding of the formation of tripartite complexes. We used the previously established optimal 1N_R_:2.5P (decamer:tetramer) stoichiometry for N_R_-P condensates [31] and triggered the reaction by adding M_2-1_ at 1:1 stoichiometry to form the known high-affinity complex with P, which yields a 1N_R_:2.5P:2.5 M_2-1_ stoichiometry for the tripartite complex. We observed that M_2-1_ is incorporated slowly into the preformed N_R_-P droplets with a maximum incorporation from 2 h onward, with an evident increase in the size of the droplets (Figure 3A). However, if we preform a 1P: 1M_2-1_ complex (under non-LLPS conditions) and trigger the reaction with N_R_, M_2-1_ is present in the condensate from its initiation (Figure 3B and Appendix A). In a separate experiment, we preformed the tripartite complex with no fluorophores at the reference stoichiometry and added small amounts of FITC labeled M_2-1_. In these experiments, we observed a slow M_2-1_ incorporation (Figure 3C), indistinguishable from the processes observed after the addition of M_2-1_ to a N_R_-P bipartite condensate (Figure 3A). The fact that the rate of formation of N_R_-P condensates is indistinguishable from the P-N_R_-M_2-1_ condensates (Figure 3A,C,D) strongly suggests that the interaction between N and P is the essential driver of the tripartite condensates, confirming findings at equilibrium [30,31]. The rate of incorporation of each of the three protein components is identical (Appendix A), suggesting that the rate limit is not related to the chemical nature of the proteins or the surface but to the physicochemical properties of the condensate; i.e., charge, viscosity, and surface tension of both bipartite and tripartite condensates. The time-dependent increase in the size of droplets with the addition of M_2-1_ prompted us to analyze the effect of the latter on the condensate. Indeed, a gradual increase in the concentration of M_2-1_ results in a marked increase in the size of the droplets, strongly suggesting a modulatory effect of M_2-1_ on the size of the condensates (Figure 3E), similar to its effect on the size of the P-M_2-1_ bipartite condensate (Figure 2B).

To confirm the formation of the different condensates in cells, we carried out transfection experiments using the A549 model cell line. When the three proteins were co-transfected, condensates containing the three proteins were observed (Figure 3F). When we transfected pairs of the components, N-M_2-1_, N-P, and P-M_2-1_, we only observed condensates in the N-P co-transfection, consistent with our previous findings (Appendix A). None of the individually transfected proteins yield condensates in cells. 

An important parameter to define the nature of the interactions involved in LLPS condensates is ionic strength. To this end, we analyzed the effect of sodium chloride on the formation of different condensates. While homotypic droplets of P only take place at salt concentrations above 100 mM, M_2-1_ shows an opposite effect, with maximum phase separation taking place at low ionic strength (Figure 4A). On the other hand, N_R_-P, P-M_2-1_, and N_R_-P-M_2-1_ heterocondensates are insensitive to salt, indicating that the overall balance of interactions is different from the homocondensates with little or no participation of electrostatic interactions in the overall stabilization of the heterocondensates (Figure 4A). Figure 4B shows the effect of salt monitored by turbidity.

To investigate the resulting protein component concentration and stoichiometries within the condensates, we carried out partition experiments measuring concentrations in the dilute and dense phases, respectively (Methods and Appendix A). At the different conditions tested, the proteins are concentrated from 5- to 50-fold with respect to their initial concentration prior to LLPS triggering. In the case of the bipartite P-M_2-1_ condensates, excess of P leads to a 20-fold concentration of P and 5-fold of M_2-1_, while excess M_2-1_ leads to a 28-fold and 14-fold concentration of P and M_2-1_, respectively (Table 1). In the case of P-N_R_ bipartite condensates, the initial ratio of 2.5:1 P:N_R_ shifts to 1:1 in the dense phase. Noticeably, the 2 P:1 M_2-1_ ratio in the dense phase holds for bipartite and tripartite condensates (Table 1). Under these experimental conditions, N_R_ concentrates the most, and shifts from a lower initial fraction of P-N (2.5:1) and P-N_R_-M_2-1_ (2.5:1:2.5) to 1:1 and 1:1:0.5 for bipartite and tripartite condensates, respectively (Table 1). 

### 3.3. M_2-1_-RNA Condensates Form Subcompartments within the In Vitro Tripartite Condensates

Previous work showed that M_2-1_ and newly synthesized RNA differentially concentrate and partition to the core of the viral factories, while P, L, and N localize at the periphery of the granules in infected cells or when the four proteins where transfected [29]. As a means to dissect this phenomenon, we therefore wanted to evaluate the possibility of in vitro condensation of M_2-1_ with RNA. We first made use of a 20-mer RNA corresponding to the gene end (GE) of the SH protein, RNA_RSV20_. A grid of varying concentrations of M_2-1_ and RNA was established, and we detected a narrow window of concentration of both components yielding condensates (Appendix A). This window corresponds to 2.5 µM of M_2-1_ and 1.25 µM RNA, with both components fluorescently labeled (Figure 5A). The process appears to build on homotypic M_2-1_ condensates that incorporate RNA at a 2 M_2-1_:1 RNA_RSV20_ ratio, where the increase in the fluorescence intensity and change in the droplet aspect suggest a modification of its properties. As the RNA concentration increases, the droplets fade and abruptly disappear at 1 M_2-1_:2 RNA_RSV20_, which is precisely the solution-binding stoichiometry as judged by EMSA (Figure 5B) and fluorescence polarization titration experiments, in line with what we described previously [37]. The fact that the saturation of the RNA-binding site dissolves heterotypic M_2-1_: RNA_RSV20_ condensates indicates that the RNA-binding site of M_2-1_ partakes in self-homotypic interactions within both types of condensates. The dissociation constant of RNA_RSV20_ is 20 nM, which indicates that it will preferentially interact with the RNA-binding site of M_2-1_ over other protein–protein or protein–RNA interactions. 

Next, we wanted to test the effects of a longer and heterogeneous RNA and, given that no sequence specificity could be defined for M_2-1_, we decided to use a consistent mixture of different RNA sequences with a defined length (70 bases) and used purified and unfolded yeast tRNA as a model of RNA ligand that may represent regions of a genome, and we refer to it as tRNA_70_. We observed heterogeneous condensates between 8:1 and 16:1 M_2-1_:tRNA ratios, different from the homotypic condensates of M_2-1_ formed at the same buffer conditions (Figure 5C). RNA condensates are often not highly regular and reflect more rigid states that with time, concentration, or other variables turn into aggregate-like forms [48]. FRAP analysis revealed that neither the M_2-1_ nor M_2-1_-tRNA_70_ condensates show significant recovery, indicative of the decreased liquid properties of the resulting condensates. However, these observations merit further investigation (Appendix A).

To determine binding and stoichiometry, we carried out an EMSA assay, monitoring ethidium bromide fluorescence of the free tRNA_70_ and the shifted position caused by the formation of the complex with M_2-1_ (Figure 5D, top). The disappearance of the free tRNA can be used as an accurate indicator of the stoichiometry and yields two M_2-1_ tetramers per tRNA molecule (Figure 5D, bottom). The binding process is cooperative, in line with our previously observed results [37]. However, we cannot determine an accurate dissociation constant under these conditions. At a 2:1 M_2-1_: tRNA_70_ binding saturation ratio, no condensate is observed, which only starts from an 8:1 ratio onward (Figure 5C), suggesting that the excess of M_2-1_ over tRNA is required, likely reflecting involvement of the homotypic M_2-1_-M_2-1_ interactions and consistent with the tendency to undergo homotypic LLPS. 

We evaluated the impact of ionic strength on both types of M_2-1_-RNA condensates and found that they are both dissolved at 150 mM NaCl (Figure 6A). The appearance of the condensates is more regular in the case of the shorter RNA, something that can be assigned to less rigid structures because of a higher valency of the tRNA. However, we observe that these different aspects of the condensates tend to be compensated with time. 

Homotypic M_2-1_ condensates are also sensitive to salt but require higher ionic strength for dissolution (Figure 6A top, taken from Figure 4A for comparison). This difference in salt sensitivity is clearly observed when turbidity is analyzed (Figure 6B). When the condensation reaction is triggered by addition of M_2-1_ to tRNA, turbidity reaches a plateau in 15 min, being fully reverted by addition of salt, suggesting that the maturation inferred from low FRAP recovery does not affect reversibility (Figure 6C). Based on the hypothesis of a strong electrostatic component, we hypothesize that a charged stabilized coacervate takes place and this would require a certain space charge distribution, i.e., an amphiphilic nature of M_2-1_. The scheme of Figure 6D represents this hypothesis where one negatively charged spot per protomer may interact with the positively charged RNA-binding domain, providing a multivalency that generates a 3D network of electrostatic interactions holding both homotypic condensates. There are two candidate regions that contain two consecutive negative charges, E70-E71 and E118-E119. In the case of RNA heterotypic condensates, the negative charge is provided by the RNA phosphate backbone regions that are not interacting with the positively charged side chains in the core RNA-binding domain.

RSV viral RNA was found within the liquid viral factories together with M_2-1_ in IBAGs, whereas P, N, and L were located at their periphery [29]. This prompted us to investigate if and how RNA incorporates into the in vitro tripartite condensates. For this, we produced condensates of P, M_2-1_, and N_R_, as described above (Figure 3A). Next, we added RNA_RSV20_ and followed the incorporation of the fluorescently labeled oligonucleotide. We observed that its incorporation is slow, starting at 90 min (Figure 7A), ultimately resulting in an inhomogeneous distribution, in which the RNA appears to concentrate inside the intrabodies toward the periphery of the droplets (Figure 7B). Due to the difficulty in labeling tRNA, we could not observe its incorporation into the condensates.

### 3.4. Bioinformatics Analysis of the Intrinsic Disorder Propensity of the RSV M_2-1_, P, and N Proteins and Their Predisposition for Phase Separation

To gain insight into the amino acid sequence-based features related to the phase separation behavior of the RSV M_2-1_, P, and N proteins, we subjected them to bioinformatics analysis aimed at finding propensities of each protein for intrinsic disorder and phase separation. The oligomeric nature of the proteins, i.e., their multivalency, is essential as a common property of condensate participating molecules, but this, combined with disordered regions, is also a hallmark of condensate formation tendencies [49].

Results of these analyses are summarized in Figure 8. For each protein, an intrinsic disorder profile generated by a set of commonly used per-residue disorder predictors is present together with the plot showing the distribution of the propensity to trigger droplet formation within their sequences (see Section 2). 

Figure 8A suggests that N is the most ordered of the three proteins analyzed in this study (~20% of its residues are predicted as disordered). As per FuzDrop platform outputs [44], the protein displays the lowest p_LLPS_ value of 0.1722 and contains one relatively short droplet-promoting region (DPR) (residues 137–157). According to this algorithm, N would be acting as a droplet client.

Figure 8B represents disorder and LLPS propensity data for P. Clearly, with the exception of the central region where the four-helix bundle tetramerization domain is located (residues 130–170), most of this protein (~85%) is predicted to be disordered, as it was previously determined both in experiments and bioinformatics [17]. P is also characterized by a very high p_LLPS_ value of 0.9793, which defines it as a droplet-driving protein capable of spontaneous LLPS. Furthermore, there are four DPRs in this protein, residues 1–14, 23–39, 44–100, and 190–241, which together cover almost 60% of its sequence. In line with previous studies, DPRs are located within the N- and C-terminal regions of P.

The antiterminator factor M_2-1_ is predicted to be rather flexible and contains significative levels of disorder according to this algorithm. In fact, most of its residues have disorder scores exceeding the threshold of 0.15, and depending on the disorder predictor used, up to 42% of the M_2-1_ residues are predicted as disordered (i.e., have disorder score ≥ 0.5, see methods). However, most of the structure appears folded with a defined secondary and tertiary structure from X-ray crystallography [11]), and this flexibility is likely reflecting undescribed dynamic features. A fully folded NMR structure of the core RNA-binding domain suggests that flexibility may lie in other regions of the protein, including a Cys3His zinc-binding domain and a C-terminal disordered stretch [14]. Based on the FuzDrop algorithm, the probability of the spontaneous LLPS for this protein, p_LLPS_, is 0.2002, which is well below the threshold of 0.6 defining proteins as LLPS drivers. However, a DPR (residues 164–194) is found at the C-terminal disordered region. Given the fact that M_2-1_ acts as both client and modulator (Figure 3), different principles, i.e., electrostatic versus other LLPS-prone characteristics, such as cation-pi, pi-pi interactions, and hydrophobic intractions, may operate for each role. Nevertheless, M_2-1_ binds with low nanomolar affinity to a 20-residue stretch of P, which would be recruiting it as a client at high ionic strength where there is no coacervation.

## 4. Discussion

It is widely accepted that most viruses investigated to date condense their gene function machinery into the membraneless liquid-like viral factories. The Mononegavirales order, in particular, was the group with the first and best-known examples of this new paradigm. This is symbolized by Negri bodies from RABV, which were known for over a century [50,51] but whose actual nature was uncovered only recently [52]. The replication complex is formed by three common proteins to this group of viruses (N, L, and P) and, in the particular case of pneumoviruses, the antiterminator factor M_2-1_. This implies a series of common themes among these viruses, proteins that differ largely in amino acid sequences but maintain structures and functions. The formation of viral condensates is one of them, and we set out to investigate the biochemical mechanistic grounds recapitulating from the pure components of RSV, focusing on P, N, and M_2-1_. 

In this work, we show that the transcription antiterminator M_2-1_ undergoes homotypic LLPS with a stronger propensity than P [31,53], even in the absence of crowding agents and in solutions with low ionic strength and against the minimal, if any, propensity predicted by the FuzDrop algorithm. The condensates appear as bead-like sticky structures, compared with the regular spherical coalescent droplets in the presence of crowder and salt (Figure 1), reminiscent of other protein–RNA condensate systems [48]. Moreover, both P and M_2-1_ mutually potentiate the formation of stoichiometrically tuned heterotypic condensation, where size is modulated by excess M_2-1_. Considering that M_2-1_ and P form a high affinity 1:1 stoichiometric complex [17] based on a 20-amino-acid binding stretch in P [47], this M_2-1_-modulated condensate points at a weak and cooperative network of interactions outside the high-affinity binding site. In fact, deletion of either the N- or C-terminal domains of P still yield heterotypic LLPS with M_2-1_, in line with our previous observations that both N- and C-terminal domains of P participate in LLPS through low amino acid sequence specificity interactions involving helical regions of both domains [31]. The fact that both proteins cannot form condensates in transfection experiments confirms that the pair N_R_-P is the primary condensate driver but is not inconsistent with M_2-1_ and P interactions in the dense phase playing key roles in the complex network of interactions that may hold and modulate viral factories.

Tripartite condensates were readily formed with M_2-1_ either as client of a bipartite N_R_-P condensate or from a preexisting stoichiometric high-affinity P-M_2-1_ complex triggered by addition of N_R_. The incorporation of M_2-1_ is very slow, taking place with a similar rate if additional M_2-1_ is added to a preformed tripartite complex. Since the high-affinity interaction rate of P and M_2-1_ in solution is on the sub-second time range, the slow incorporation to the condensate suggests a physical and energetic barrier yet to be uncovered. However, since P and N_R_ are incorporated with a similarly slow rate and given that their surface and chemical properties are different, the barrier appears to be physical. We observe that increasing amounts of M_2-1_ modulate the size of the tripartite complex in a similar manner to the P-M_2-1_ bipartite condensates, strongly suggesting that the size of the condensates in the context of the viral factories is modulated by the stoichiometric interplay between P and M_2-1_. 

Prompted by the fact that M_2-1_ was observed concentrated at the center of the RSV viral factories together with RNA, we pursued the possibility of the formation of protein–RNA condensates. Indeed, such condensates were formed at salt concentrations below 100 mM and were different in the morphology and regularity of the shape compared to those of the homotypic M_2-1_ condensates. This behavior is reminiscent of the behavior of protein–RNA condensates, which were shown to form sticky beadlike condensates with a decreasing fluidity as they develop [48]. Both homotypic and tRNA heterotypic M_2-1_ condensates show a slow and small fraction of recovery, indicative of loss in the fluidity. However, the condensates are reversed by simple addition of salt, suggesting that they do not represent a kind of rigid amorphous or fibrous aggregates. Our working hypothesis is that M_2-1_ is an amphiphilic protein capable of forming electrostatic-based self-interactions between the negatively charged regions and the positive-RNA-binding site, which is an absolute requirement for LLPS. Over-stoichiometric excess of RNA dissolves the condensates, indicating that RNA-saturated M_2-1_ cannot phase separate and suggests a fine condensation modulation by the RNA:M_2-1_ ratio that is biochemically compatible with the viral factory context. The fact that condensation requires excess M_2-1_ is indicative of homotypic interactions coexisting in the context of heterotypic LLPS and likely involves other regions of M_2-1_ interacting with the RNA-binding site. Furthermore, our results strongly suggest minimal if any base sequence preferences for condensation, which is reasonable if one considers that heterogeneous sequences and RNA lengths will be the product of viral genome transcription with a large variety of sequences. Overall, M_2-1_-RNA condensation appears to be driven by an electrostatic-dependent complex coacervation as was previously described [45,54]. The mechanism corresponds to the reentrant phase transitions, which can be tuned by RNA, modulate layered functional topologies, and enable biochemical timekeeping of self-limiting reactions (reviewed in [55]). In any case, in this particular protein–RNA condensate, M_2-1_ appears to be the main driver. The short RNA oligo corresponding to the GE regulatory sequence of one of the RSV proteins (SH), RNA_RSV20_, is incorporated to the tripartite condensates at a very slow rate similar to the chemically different proteins. This merits further investigation but suggests a barrier that can be attributed to condensate surface properties, condensate density, and high phase viscosity, or both. Addition of RNA_RSV20_ leads to the formation of heterogeneities in the form of irregular granules within the droplets. We speculate that these are governed by the RNA compartment concentration equivalent to length increase product of transcription. This is consistent with the picture of a liquid viral factory assembly being modulated by macromolecular and ligand trafficking and ultimately leading to single genome nucleocapsids [27]. In particular, M_2-1_-RNA anisotropic condensates are likely to correlate with the formation of granular structures (IBAGs) described to take place within the RSV viral factories [29,56].

Several important conclusions can be drawn from the ionic strength dependence of the different condensates considered in this study. First, P and M_2-1_ show an opposed behavior, suggesting not only that ionic interactions are not required for homotypic LLPS of P but intramolecular electrostatic interactions can down-modulate the process [31]. Interestingly, P-M_2-1_ condensates are insensitive to ionic strength; this can in principle result from the opposed effect of salt or that non-ionic interactions hold these bipartite condensates. However, N_R_-P and tripartite N_R_-P-M_2-1_ condensates are also insensitive to ionic strength. Therefore, ionic strength impacts mostly on M_2-1_-RNA condensate heterotypic interactions and the M_2-1_ homotypic interactions within those condensates. 

A known feature among Mononegavirales is the fact that N and P genes are located at the 3′ end of the genomes, and the protein abundance is to a great extent determined by transcription polarity [1,57]. It is safe to hypothesize that N and P are the most abundant replication proteins and start building up at earlier stages; after the initial round of transcription, M_2-1_ will increase at later stages, and the polymerase L will display the lower levels. In addition, N_R_-P were shown to be sufficient for condensation in vitro and in cells [15,16,31]. After an initial buildup of small condensates, which increase in size as they are either expressed as transfected or under infection [31], M_2-1_ could be incorporated into growing viral factory condensates as a client of P. It remains to be established whether M_2-1_ can be incorporated as a P- M_2-1_ complex or only as a client of P. The RSV polymerase L is tightly bound and stabilized by P [20,21], and we can assume it to be bound to P throughout the infection cycle, so it can be considered a client of P from its very synthesis, since P will be always present in excess. Moreover, L does not need to build up, since it is an enzyme that will be active even in small amounts and is largely potentiated by the extreme macromolecule and substrate concentration within a viral replication and transcription condensate [58]. As RNA transcription levels within the factories are increased by the gradual condensation as the infection progresses, it may form heterotypic condensates with M_2-1_, described as the subcompartment-termed IBAGs in the context of infection [29] (Figure 9). An important phenomenon observed in our work is the strong size-modulatory effect of M_2-1_, particularly noticeable in the excess of the protein over the population present within the tripartite N_R_-P-M_2-1_ condensates, establishing M_2-1_ as both client and size modulator. Recent work uncovered compounds that can harden RSV viral factories by targeting M_2-1_, although it is not yet clear how the mechanism operates [59]. In any case, M_2-1_ plays an important role in line with the modulatory effect we observe here. The present work recapitulates the biochemical mechanism through which the main acknowledged viral players behind RSV factory assembly operates. In doing so, it provides key steps that can be addressed by reverse genetics and exposes essential modulatory reactions that can be targeted in the context of a novel LLPS-based platform for antiviral discovery. Future studies within both the infection context and in vitro will link the role of the antiterminator in transcription processivity with that modulating the viral factory condensation.

## Figures and Tables

**Figure 1 viruses-15-01329-f001:**
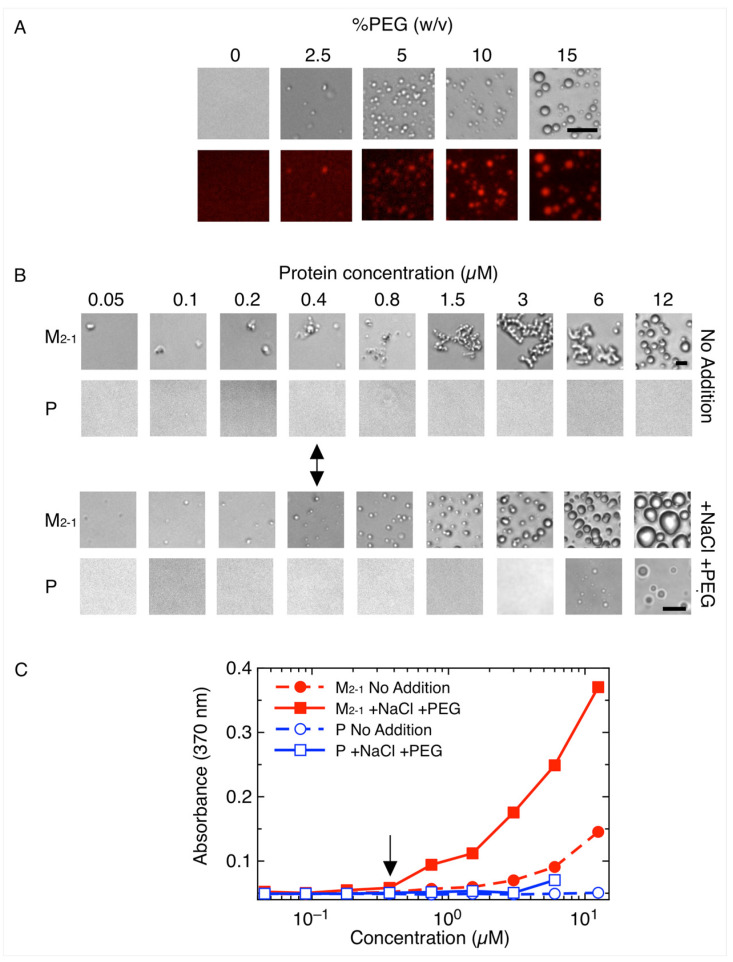
Homotypic LLPS of M_2-1_ compared to P. (**A**) Effect of crowding agent on homotypic LLPS; 5 µM M_2-1_ was incubated in the presence of increasing concentrations of PEG-4000. Scale bar = 10 µm. (**B**) P and M_2-1_ were tested at concentrations ranging from 0.05 to 12.5 µM in the presence or absence of 150 mM NaCl and 15% PEG-4000. Arrow indicates the saturation concentration of M_2-1_ in both conditions. Scale bar = 10 µm. (**C**) Effect of protein concentration on homotypic LLPS monitored by turbidity.

**Figure 2 viruses-15-01329-f002:**
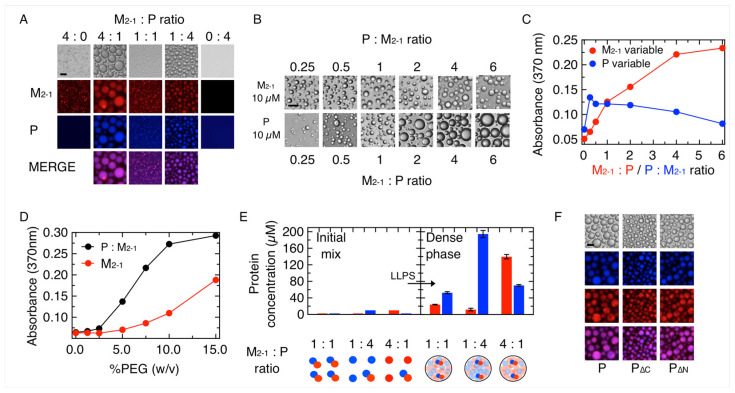
Heterotypic condensation of P and M_2-1_. (**A**) P and M_2-1_ were incubated at different ratios (protein concentration ranged from 2.5 to 10 µM) and visualized by brightfield and fluorescence microscopy. Scale bar = 10 µm. (**B**) Bright field microscopy of samples containing fixed concentration of M_2-1_ (10 µM) and varying concentrations of P (upper panel) and fixed concentration of P (10 µM) and varying concentration of M_2-1_ (lower panel). Scale bar = 10 µm. (**C**) Turbidity assay monitoring absorbance at 370 nm of the samples analyzed in (**B**). (**D**) Effect of crowding agent (PEG-4000) concentration on homotypic M_2-1_ vs. heterotypic P-M_2-1_ condensation monitored by turbidity. (**E**) Top, initial protein concentration vs. dense phase concentration of P (blue) and M_2-1_ (red) in a partition experiment (see Section 2) at different ratios of heterotypic P-M_2-1_ condensates (*n* = 3 ± s. e.). Bottom, schematic representation of the initial and dense phase conditions with their respective stoichiometry. Solid spheres refer to proteins in solution and stoichiometric complex formation. Translucent spheres refer to proteins within the condensate. (**F**) Effect of P and its deletions P_∆N_ and P_∆C_ on the formation of heterotypic condensates with M_2-1_ in a 4 M_2-1_:1 P ratio. Scale bar = 10 µm.

**Figure 3 viruses-15-01329-f003:**
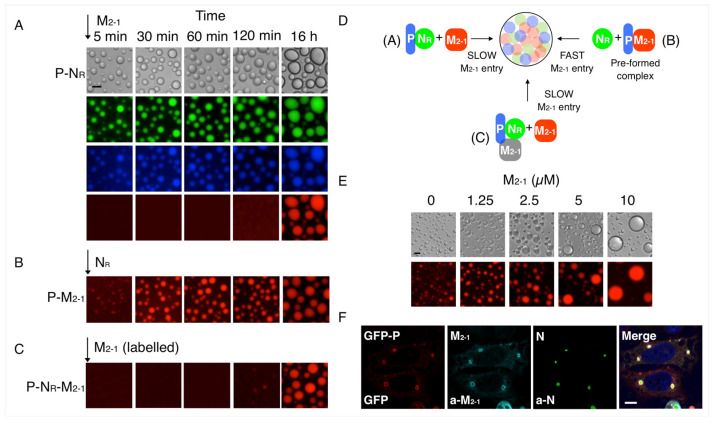
Tripartite P-N_R_-M_2-1_ condensates and incorporation of M_2-1_. (**A**) Incorporation of M_2-1_ over time to bipartite condensates of P-N_R_. Preformed P-N_R_ condensates were incubated for 1 h followed by addition of M_2-1_ (arrow). N is shown in green, P in blue, and M_2-1_ in red. Scale bar = 10 µm. (**B**) Formation of tripartite heterotypic condensates triggered by N_R_ (see Appendix A). Preformed P-M_2-1_ soluble complex incubated for 10 min and then N_R_ was added (arrow). (**C**) Incorporation of M_2-1_-FITC to tripartite condensates of unlabeled P-N_R_-M_2-1_ (see Appendix A). Preformed P-N_R_-M_2-1_ tripartite condensates were incubated for 1 h followed by addition of M_2-1_-FITC (arrow). (**D**) Schematic representation of the incorporation of M_2-1_ to tripartite condensates. (**E**) Effect of increasing M_2-1_ protein concentration on P-N_R_-M_2-1_ tripartite condensates. Scale bar = 10 µm. (**F**) Co-localization of RSV proteins in transfected cells. A549 cells were co-transfected with plasmids encoding GFP-P, M_2-1_, and N. After 24 h, the proteins were detected by direct GFP fluorescence or with anti-M_2-1_ or anti-N antibodies by IFI. Nuclei were stained with DAPI. Scale bar = 10 µm.

**Figure 4 viruses-15-01329-f004:**
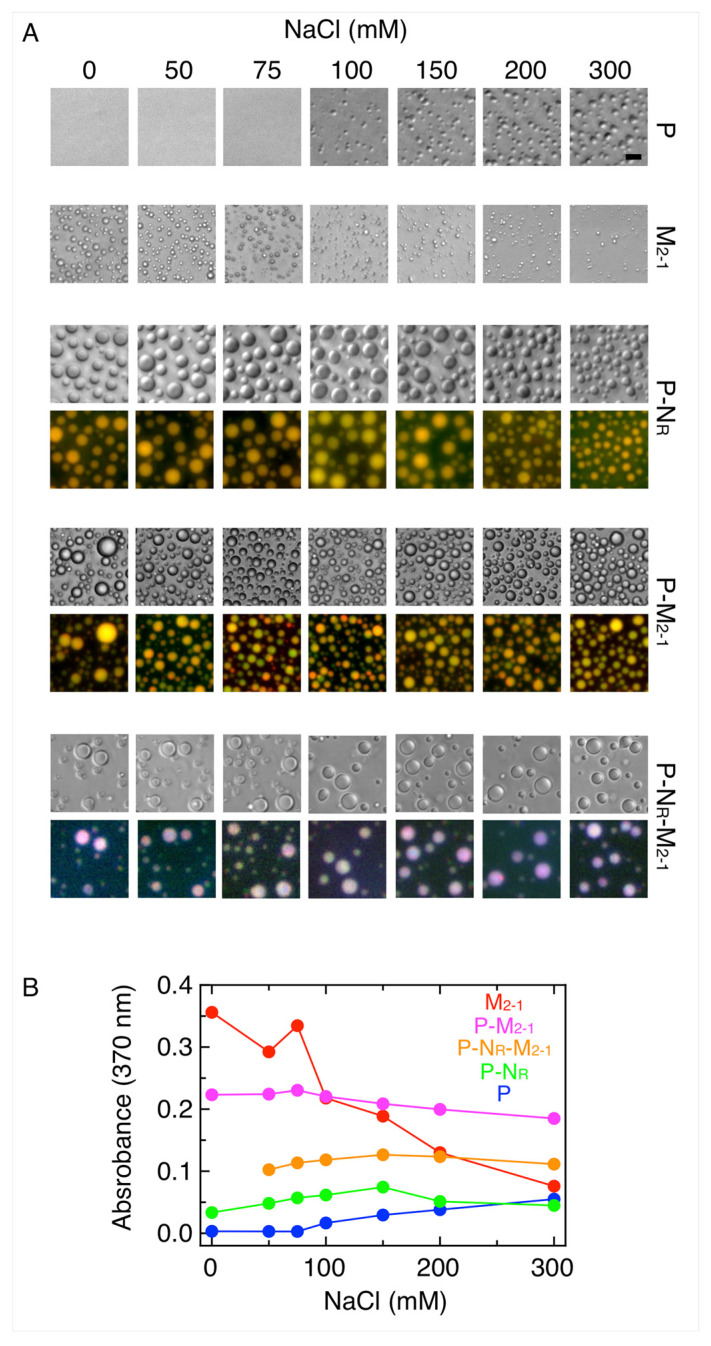
Effect of ionic strength on homotypic and heterotypic condensates. (**A**) Homotypic condensates of P (5 µM) and M_2-1_ (5 µM) and heterotypic condensates of P-N_R_ (1.25 µM and 0.5 µM, respectively), P-M_2-1_ (1.25 µM and 5 µM, respectively) and P-N_R_-M_2-1_ (1.25 µM, 0.5 µM, and 1.25 µM, respectively) at increasing concentration of NaCl. Scale bar = 10 µm. (**B**) Turbidity assay of samples from (**A**) monitored by absorbance signal at 370 nm.

**Figure 5 viruses-15-01329-f005:**
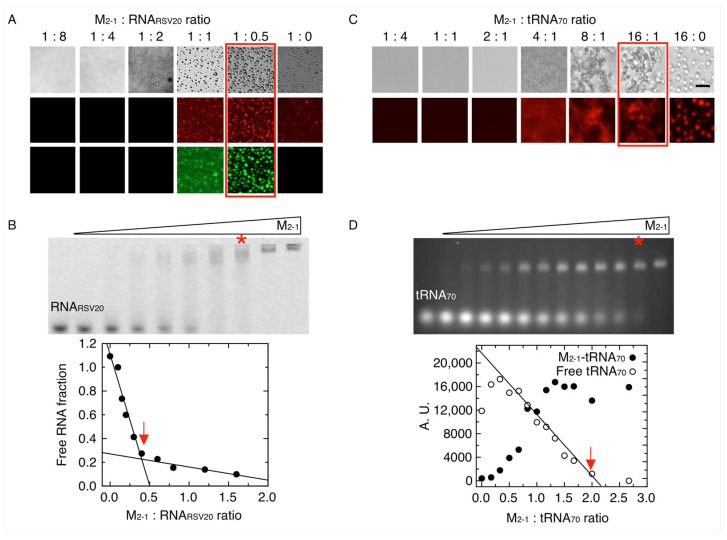
Modulation of heterotypic M_2-1_-RNA condensates by binding stoichiometry. (**A**) Samples with fixed concentration of Cy5-M_2-1_ (2.5 µM) were incubated with varying concentrations of FITC-RNA_RSV20_ ratio. A maximum condensation effect is seen at a 1:0.5 ratio (highlighted in red) in 25 mM HEPES pH 7.5, 5% PEG and 100 mM NaCl buffer. (**B**) Top, electrophoretic shift mobility assay (EMSA) of stoichiometric complex M_2-1_-RNA_RSV20_ formation varying the concentration of M_2-1_ (0 to 320 nM) with fixed concentration of RNA_RSV20_ (200 nM). The asterisk refers to the stoichiometric complex ratio formation. Bottom, plot represents the fraction of free RNA densitometry from the EMSA as a function of M_2-1_:RNA_RSV20_ ratio. Arrow indicates the solution binding stoichiometry. (**C**) Samples with fixed concentration of M_2-1_ (2.5 µM) were incubated with varying concentrations of tRNA_70_ in heterotypic condensates. A maximum condensation effect takes place at a 16:1 ratio (highlighted in red) in 25 mM HEPES pH 7.5, 5% PEG and minimum NaCl buffer. Scale bar = 10 µm. (**D**) Top, EMSA of M_2-1_-tRNA_70_ stoichiometric complexes formation varying the concentration of M_2-1_ (0 to 4 µM) with fixed concentration of tRNA_70_ (1.5 µM). The asterisk refers to the stoichiometric complex ratio formation. Bottom, plot depicts free M_2-1_-tRNA_70_ complex (full circles) and RNA (unfilled circles) densitometry from the EMSA assay. Arrow indicates the solution-binding stoichiometry.

**Figure 6 viruses-15-01329-f006:**
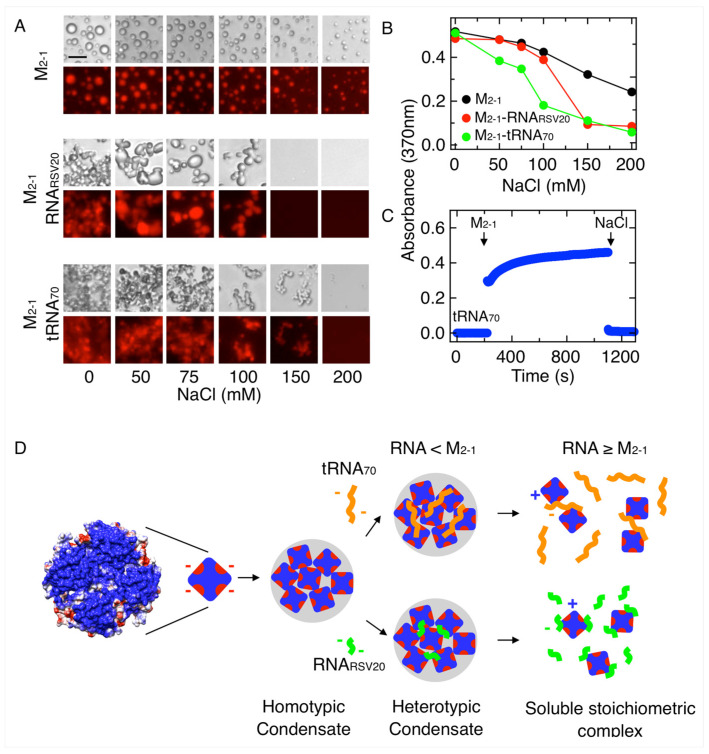
Ionic strength dependence and reversibility of M_2-1_-RNA condensates. (**A**) Effect of NaCl on M_2-1_ (5 µM) homotypic and heterotypic, and M_2-1_-RNA_RSV20_ (2 M_2-1_:1 RNA_RSV20_ ratio) and M_2-1_-tRNA_70_ (16 M_2-1_:1 tRNA_70_ ratio) heterotypic condensates. Scale bar = 10 µm. (**B**) Turbidity measurement of samples from (**A**) monitored by absorbance at 370 nm. (**C**) Turbidity kinetic assay monitoring heterotypic condensation triggered by addition of 5 µM M_2-1_ to a solution of 0.31 µM tRNA_70_ (16 M_2-1_:1 tRNA_70_ ratio) and reversed by increasing NaCl to 0.5 M. (**D**) Coulombic surface scheme of M_2-1_ structure. The red areas correspond to negatively charged regions (E70-E71 and E118-E119) and the blue ones to positively charged regions. Scheme of interactions of M_2-1_ in homotypic condensates and interactions of M_2-1_ with RNA both in heterotypic condensates and in soluble stoichiometric complex. The formation of homotypic M_2-1_ condensates is affected by the addition of sub stoichiometric short (20 bases) or long (70 bases) RNA. These heterotypic condensates are irregular but do not correspond to amorphous aggregates. RNA excess dissolves heterotypic condensates.

**Figure 7 viruses-15-01329-f007:**
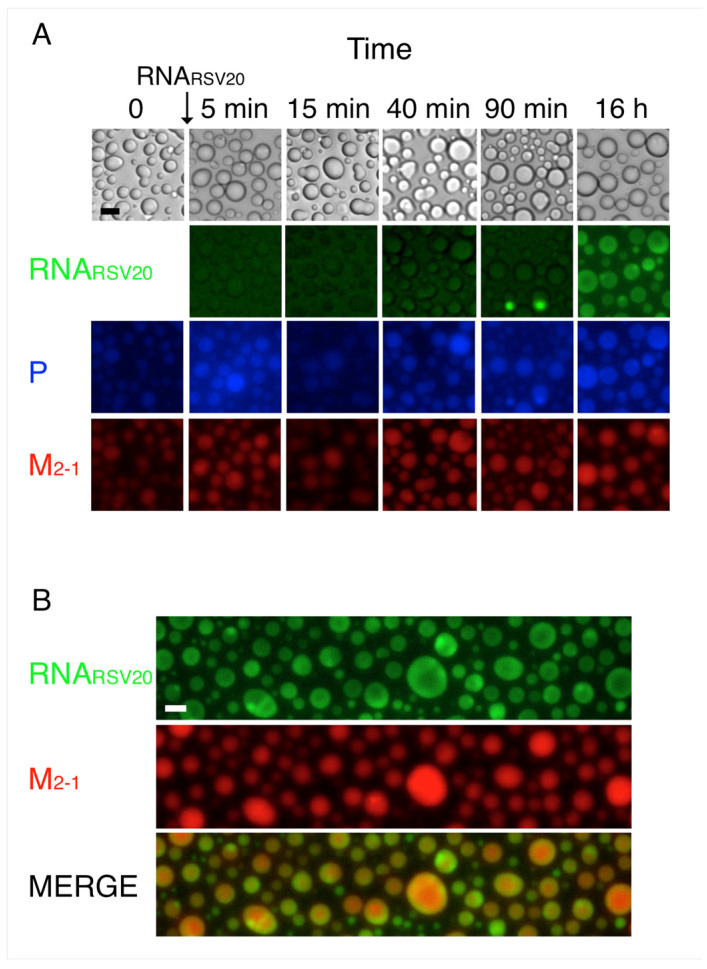
P-N_R_-M_2-1_ tripartite heterotypic condensates with RNA_RSV20_. (**A**) Effect of RNA_RSV20_ on tripartite condensates of P-N_R_-M_2-1_ over time. 0.625 µM RNA_RSV20_ was added to preformed tripartite condensates at a 2 M_2-1_:1 RNA_RSV20_ ratio in 25 mM HEPES pH 7.5, 5% PEG, and 75 mM NaCl buffer. (**B**) Representative fluorescence microscopy images of samples from (**A**) after overnight incubation. Heterogeneous distribution of RNA can be seen within the condensate. Scale bar = 10 µm.

**Figure 8 viruses-15-01329-f008:**
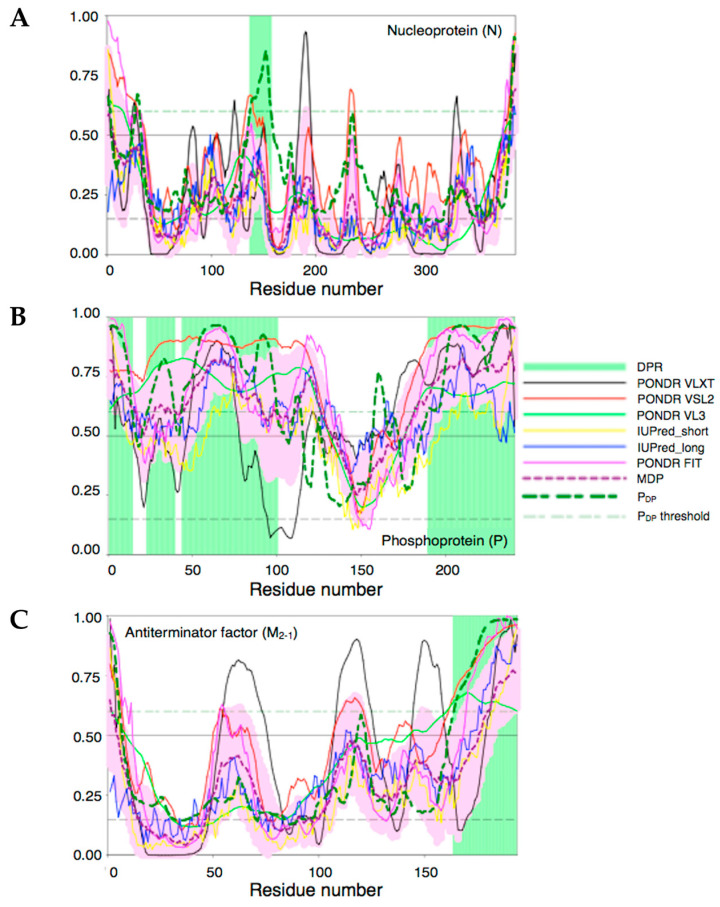
Intrinsic disorder and LLPS propensity of the RSV N (**A**), P (**B**), and M_2-1_ proteins (**C**).

**Figure 9 viruses-15-01329-f009:**
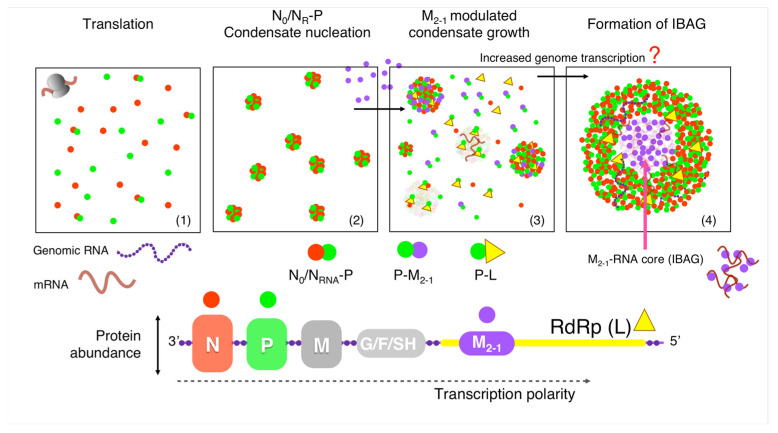
Mechanistic hypothesis for viral factory condensation based on our in vitro analysis. Proteins N and P are the first ones to be synthesized and in large quantities (1) and are known to be the main drivers for condensation (2). As M_2-1_ levels increase at later stages of the infection cycle, it incorporates into the condensate nuclei to increase the size of the condensates (3). We propose that as polymerase is synthesized at a later stage and low amounts, it is tightly bound to P and is incorporated to condensates as clients (3). Although not known at this stage, we hypothesize that, exacerbated by the high concentrations of both enzyme and template, genomic transcription drastically increases and the increased mRNA transcripts recruit M_2-1_ from the protein phase to the IBAGs subcompartments (4). NS1 and NS2 genes are upstream N and not shown on this schematic representation of the viral genome. The same goes for M_2-2_, the product of the expression of the second ORF of the viral gene M2.

**Table 1 viruses-15-01329-t001:** Initial and dense phase stoichiometry and fold concentration of proteins in heterotypic condensates after partition experiments.

	Stoichiometry		Fold Concentration ^1^
	Initial	Dense Phase	
P:M_2-1_	04:01	16:01	19.5:4.8
P:M_2-1_	01:01	01:00.5	21.2:9.6
P:M_2-1_	01:04	01:02	28:14:00
P:N_R_	2.5:1	01:01	19.7:40.7
P:N_R_:M_2-1_	2.5:1:2.5	01:00.5	24.2:46.7:12.9

^1^ From concentration values determined in Appendix A.

## Data Availability

Data is contained within the article or Appendix A.

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
