# Peer review of "Assembly of the Tripartite and RNA Condensates of the Respiratory Syncytial Virus Factory Proteins In Vitro: Role of the Transcription Antiterminator M2-1"

_viruses, 2023, doi:10.3390/v15061329_

Round 1
Reviewer 1 Report
General comments
This paper is in line with previous work performed on the study of RSV viral factories (VF) also called “cytoplasmic inclusion bodies” (IBs) in infected cells or in vitro. It was shown previously that these VF form by a liquid-liquid phase separation (LLPS) mechanism and that the scaffold of these VF is made by viral P:N-RNA complexes. It was also determined previously that VF are complex and contain sub-compartments called IBAGs that are made of complexes between the RSV M2-1 protein and viral mRNAs. In this work, the authors have used a biochemical approach to dissect the formation and properties of pseudo-VF made in vitro using recombinant proteins. They show for the first time that, in contrast to P, M2-1 can easily form homotypic condensates; M2-1 is less dependent on crowding agents and salt for homotypic LLPS, compared to P. They studied the effect of P:M2-1 ratios on formation of heterotypic condensates as well as tripartite complexes (N-P-M2-1), using different combinations of preformed complexes and ratios. A preformed M2-1-P complex is required for rapid incorporation of M2-1 in IBs. They show that M2-1 homo-condensates are highly modulated by RNA and M2-1:RNA heterocomplexes; they show that an external RNA can incorporate into pseudo-IBs formed by P:M2-1:N-RNA. These results shed light on the variability of homo- and hetero-complexes that can form in RSV-infected cells between N, P and M2-1 proteins and RNA, and that can only be revealed by in vitro experiments where biochemical/biophysical conditions can be modulated and finely tuned. I only regret that the authors did not test an M2-1 phosphomimetic variant, or in vitro phosphorylated, as M2-1 function is modulated by phosphorylation.
Abstract
Line 15 : the SRV --> RSV
Line 16 : « ...and maximize transcriptase processivity”: M2-1 action is specific of transcriptase
Line 42: ...human metapneumovirus (MPNV): generally, the abbreviation used is HMPV
Introduction
Lines 53-54: “Unique to pneumovirus, M2-1 is an additional cofactor required for efficient mRNA transcription.” and the following sentences about M2-1: this is true for RSV but not for HMPV. Although important in vivo, M2-1 can be deleted from the HMPV genome that still can replicate in cultured cells, although attenuated (see Buchholz et al., 2005, JVI doi:10.1128/JVI.79.11.6588–6597.2005). For minigenome transcription assays, HMPV M2-1 is not required, while for RSV it is critical. So, I think we have to distinguish roles and functions of RSV M2-1 and HMPV M2-1. Also true for lines 540-542.
Mat & Met
Paragraph 2.8: the authors should explain how were fused the M2-1, P and N to GFP: fusion to GFP is not neutral and depends on how it is done; for example, fusing GFP to the C-terminus of P should block the formation of Nr-P complexes.
Results
Lines 303-304: the authors should remind us what are the deleted residues for P∆N and P∆C for better clarity. They should also remind us the exact position of the previously identified M2-1 nM binding site (aa 93-110)
Fig.3: the authors should indicate in the legend what means “ON” (overnight). In (F) a scale bar is missing.
For parts 3.1 and 3.2, the formation of heterocomplexes was performed with a NaCl concentration of 75 mM, as described in 2.7. However, the effect of salt on heterocomplexes was investigated latter in the article; the authors should explain at the beginning of paragraph 3.2 why they used 75mM for all these experiments.
Part 3.3 what was the NaCl concentration used for the first experiments (complexes between M2-1, and RNAs)? It is again only investigated after
I noticed that condensates between M2-1 and RNA20 and M2-1 and tRNA are different, those made with tRNA looking more aggregated (Fig.5 and 6); the authors should comment this in the Results section.
Fig.7: what was the NaCl concentration used? Again, based on experiments described above, this is critical.
Also, a control with P:N-RNA complexes and an added RNA20 without M2-1 is missing to demonstrate that the incorporation of RNA20 into IBs is mediated by M2-1 only.
In paragraph 3.4 the authors indicate that intrinsically disordered regions are the main drivers of LLPS. But they ignore the oligomeric nature of these partners, which is also recognized as an important factor, even if when proteins are well folded and poorly disordered.
Figure 9: this model does not integrate the result described in Fig.3 showing that P-M2-1 complex is likely to be incorporated in IBs, not M2-1 alone, and which is in agreement with previous published results (Richard et al., ref 27).
It also excludes the phosphorylation modulation of M2-1 which is dephosphorylated by PP1 when binding to P-PP1 complexes (Richard et al., ref 27) and should play a role for the balance between M2-1:P and M2-1:RNA complexes.
There is a typing error: NH should be replaced by SH
Should be added that for RSV NS1 and NS2 genes are upstream N and not shown on this schematic representation of the viral genome, as well as M2-2
Discussion
Lines 624-625: “A known feature among mononegavirales is the fact that N and P proteins are located at the 3’ end of the genomes,”: change “proteins” by “genes” .
Lines 631-632: “...M2-1 will be incorporated to growing viral factory condensates as client of P.”: what about preformed M2-1:P complexes?
The paper is well written although some parts are written in the preterit and others in the present tense (ex.: lines 334-336, 434, 483).
Author Response
We are glad to learn that the paper was positively evaluated by the reviewers. They made sharp and useful comments that will improve the manuscript and I shall thank them for that.
We addressed all comments and we clarify the modifications after every comment, and changes are highlighted in bold in the new manuscript
Abstract
Line 15 : the SRV --> RSV
Line 16 : « ...and maximize transcriptase processivity”: M2-1 action is specific of transcriptase
Line 42: ...human metapneumovirus (MPNV): generally, the abbreviation used is HMPV
Corrected
Introduction
Lines 53-54: “Unique to pneumovirus, M2-1 is an additional cofactor required for efficient mRNA transcription.” and the following sentences about M2-1: this is true for RSV but not for HMPV. Although important in vivo, M2-1 can be deleted from the HMPV genome that still can replicate in cultured cells, although attenuated (see Buchholz et al., 2005, JVI doi:10.1128/JVI.79.11.6588–6597.2005). For minigenome transcription assays, HMPV M2-1 is not required, while for RSV it is critical. So, I think we have to distinguish roles and functions of RSV M2-1 and HMPV M2-1. Also true for lines 540-542.
The reviewer is correct, we added the sentence “Although important in vivo, M2-1 can be deleted from the HMPV genome that still can replicate in cultured cells, although attenuated”.
Mat & Met
Paragraph 2.8: the authors should explain how were fused the M2-1, P and N to GFP: fusion to GFP is not neutral and depends on how it is done; for example, fusing GFP to the C-terminus of P should block the formation of Nr-P complexes.
Requested information added.
Results
Lines 303-304: the authors should remind us what are the deleted residues for P∆N and P∆C for better clarity. They should also remind us the exact position of the previously identified M2-1 nM binding site (aa 93-110).
Clarification added
Fig.3: the authors should indicate in the legend what means “ON” (overnight). In (F) a scale bar is missing.
Corrected
For parts 3.1 and 3.2, the formation of heterocomplexes was performed with a NaCl concentration of 75 mM, as described in 2.7. However, the effect of salt on heterocomplexes was investigated latter in the article; the authors should explain at the beginning of paragraph 3.2 why they used 75mM for all these experiments.
We added the following sentence in section 3.1 “Based on the results of Figure 1B, we used an intermediate value of 75 mM NaCl for this experiments “
Part 3.3 what was the NaCl concentration used for the first experiments (complexes between M2-1, and RNAs)? It is again only investigated after.
Added in legend for figure 5.
I noticed that condensates between M2-1 and RNA20 and M2-1 and tRNA are different, those made with tRNA looking more aggregated (Fig.5 and 6); the authors should comment this in the Results section.
We added the following sentence: “The appearance of the condensates is more regular in the case of the shorter RNA, something that can be assigned to less rigid structures because of a higher valency of the tRNA. However, we observe that these different aspects of the condensates tend to be compensated with time (not shown).”
Fig.7: what was the NaCl concentration used? Again, based on experiments described above, this is critical.
Added in the figure legend.
Also, a control with P:N-RNA complexes and an added RNA20 without M2-1 is missing to demonstrate that the incorporation of RNA20 into IBs is mediated by M2-1 only.
We did not state that M2-1 mediates entrance of RNA, not even as hypothesis. Thus, we believe this control is not necessary
In paragraph 3.4 the authors indicate that intrinsically disordered regions are the main drivers of LLPS. But they ignore the oligomeric nature of these partners, which is also recognized as an important factor, even if when proteins are well folded and poorly disordered.
To address this, we added the following text “The oligomeric nature of the proteins, i.e., its multivalency is essential as a common property of condensate participating molecules, but this combined with disordered regions are also a hallmark of condensate formation tendencies”.
Figure 9: this model does not integrate the result described in Fig.3 showing that P-M2-1 complex is likely to be incorporated in IBs, not M2-1 alone, and which is in agreement with previous published results (Richard et al., ref 27).
We corrected the model by adding M2-1-P complexes but we also believe that M2-1 can be in excess of P and thus incorporated as uncomplexed.
It also excludes the phosphorylation modulation of M2-1 which is dephosphorylated by PP1 when binding to P-PP1 complexes (Richard et al., ref 27) and should play a role for the balance between M2-1:P and M2-1:RNA complexes.
This comment is accurate and we are currently addressing the phosphorylation issue in a comprehensive way (responding to a general comment by this reviewer). However, we feel this will feed too much information into the scheme, not necessarily clarifying.
There is a typing error: NH should be replaced by SH.
Modified.
Should be added that for RSV NS1 and NS2 genes are upstream N and not shown on this schematic representation of the viral genome, as well as M2-2.
Corrected.
Discussion
Lines 624-625: “A known feature among mononegavirales is the fact that N and P proteins are located at the 3’ end of the genomes,”: change “proteins” by “genes” .
Corrected.
Lines 631-632: “...M2-1 will be incorporated to growing viral factory condensates as client of P.”: what about preformed M2-1:P complexes?
Modified: “After an initial buildup of small condensates, which increase in size as they are either expressed as transfected or under infection, M2-1 could be incorporated to growing viral factory condensates as client of P. It remains to be established whether M2-1 can be incorporated as P- M2-1 complex or only as client of P.”
Comments on the Quality of English Language
The paper is well written although some parts are written in the preterit and others in the present tense (ex.: lines 334-336, 434, 483).
Corrected
Reviewer 2 Report
Please see comments in file.

Author Response
We are glad to learn that the paper was positively evaluated by the reviewers. They made sharp and useful comments that will improve the manuscript and I shall thank them for that.
We addressed all comments and we clarify the modifications after every comment, and changes are highlighted in bold in the new manuscript
- I found few spelling mistakes frequently in the text. Please correct them.
Checked
- In Figure 1(b) top panel, phase separation of M2-1 alone with increasing concentration, the condensates look like fractal structures which turn into more spherical/liquidy condensates at highest concentrations. Could authors comment on this behavior.
An answer for this requires substantial specific experimental evidence. We added the following sentence: “This appears to respond to different physical properties of the condensates and require further investigation ”
- In Figure 1(b) bottom panel, addition of salt and PEG seems to lead to formation of more spherical condensates as well, while decreasing phase-separation concentration. Could authors dissect separately the influence of salt and PEG.
The contribution of salt at a fixed PEG concentration was carried out and is shown in figure 4.
- In M2-1 and P co-condensation, Figure 2 (A), can authors explore other ratios besides the three tested. Also, what is the net charge on P protein and M2-1 protein. Usually, the biggest condensates are made at charge balanced condition in the solution mixture.
A curve at different ratios was performed and is shown in panels B and C of the same figure.
- The authors have posited electrostatic interactions at play in M2-1 homotypic and M2-1 – RNA condensation. While some LLPS interaction for condensation of P. Can the authors provide insight towards the residues/stickers they hypothesize to be responsible for the interactions driving phase-separation in all cases, as authors have done both thorough in- vitro experiments and bioinformatic analysis.
We added the following sentence “There are two candidate regions that contain two consecutive negative charges, E70-E71 and E118-E119.”
6.Figure 2 (D), I do not understand how PEG data is adding to the story?
As we explained in the text the comparative effect of a crowder such as PEG is a further indicative of LLPS tendency. Crowders are used in most condensates studies in vitro as likely in cell concentrations are in the tens of milligrams per mL, not easily attained in vitro with recombinant proteins, particularly for a large number of experiments as those described in this paper.
- In line 300-301, the authors talk about solvent role of P in the homotypic condensates. For
the uninitiated reader it is hard to grasp, can the authors explain elaborately.
We clarified this important point raised by the reviewer by adding the following text: “Homotypic LLPS of macromolecules arise in part from these having more binding propensity for themselves than for the actual solvent molecules, so scaffolds that are often in excess are acting in fact as solvents of the other components ”
- In line 305, the authors claim that P only condensates and condensates with deletion constructs are indistinguishable. The PdeltaC condensates seem slightly smaller if the scale on the images are the same, possibly due to decrease in monomer length. The authors could change indistinguishable to another word that best describes the results.
We changed “indistinguishable” for “resemble”.
- Figure 5C, the structures of M2-1 with tRNA look more like aggregates than spherical
condensates. Please comment.
We added the following sentence to address the reviewer’s concern: “RNA condensates are often not highly regular and reflects more rigid states that with time, concentration, or other variables turn into aggregate-like forms”
- In line 529, authors talk about electrostatics VS. LLPS prone characteristics. Electrostatics
comes under the umbrella of LLPS interactions, others being cation-pi interactions, pi-pi interactions, hydrophobic interactions etc. Can the authors elaborate what precisely they mean by LLPS prone characteristics?
We addressed the reviewer’s insightful comment by modifying the sentence to “Given the fact that M2-1 acts as both client and modulator (Figure 3), different principles, i.e. electrostatic versus other LLPS prone characteristics such as cation-pi, pi-pi interactions, hydrophobic interactions, may operate for each role.”
Round 2
Reviewer 1 Report
The authors have now addressed all my comments